# Long-term risk of death and recurrent cardiovascular events following acute coronary syndromes

Pishoy Gouda[1], Anamaria Savu[2], Kevin R. Bainey[1,2], Padma Kaul[1,2,3], Robert C. Welsh [1,2]*

1 Division of Cardiology, Department of Medicine, Mazankowksi Alberta Heart Institute and University of Alberta, Edmonton, Alberta, Canada, 2 Canadian VIGOUR Centre, University of Alberta, Edmonton, Alberta, Canada, 3 Canadian Institutes of Health Research Chair in Sex and Gender Science, Edmonton, Alberta, Canada

* Robert.Welsh@ahs.ca

**Data Availability Statement:** Data cannot be shared publicly because as it was provided by the Government of Alberta under the terms of a research agreement stipulating that we do not publicly share the data. Data are available by

## Abstract

Estimates of the risk of recurrent cardiovascular events (residual risk) among patients with acute coronary syndromes have largely been based on clinical trial populations. Our objective was to estimate the residual risk associated with common comorbidities in a large, unselected, population-based cohort of acute coronary syndrome patients. 31,056 ACS patients (49.5%—non-ST segment elevation myocardial infarction [NSTEMI], 34.0%—ST segment elevation myocardial infarction [STEMI] and 16.5%—unstable angina [UA]) hospitalised in Alberta between April 2010 and March 2016 were included. The primary composite outcome was major adverse cardiovascular events (MACE) including: death, stroke or recurrent myocardial infarction. The secondary outcome was death from any cause. Cox-proportional hazard models were used to identify the impact of ACS type and commonly observed comorbidities (heart failure, hypertension, peripheral vascular disease, renal disease, cerebrovascular disease and diabetes). At 3.0 +/- 3.7 years, rates of MACE were highest in the NSTEMI population followed by STEMI and UA (3.58, 2.41 and 1.68 per 10,000 person years respectively). Mortality was also highest in the NSTEMI population followed by STEMI and UA (2.23, 1.38 and 0.95 per 10,000 person years respectively). Increased burden of comorbidities was associated with an increased risk of MACE, most prominently seen with heart failure (adjusted HR 1.83; 95% CI 1.73–1.93), renal disease (adjusted HR 1.52; 95% CI 1.40–1.65) and diabetes (adjusted HR 1.51; 95% CI 1.44–1.59). The cumulative presence of each of examined comorbidities was associated with an incremental increase in the rate of MACE ranging from 1.7 to 9.98 per 10,000 person years. Rates of secondary prevention medications at discharge were high including: statin (89.5%), angiotensin converting enzyme inhibitor/angiotensin receptor blocker (84.1%) and beta-blockers (85.9%). Residual cardiovascular risk following an acute coronary syndrome remains high despite advances in secondary prevention. A higher burden of comorbidities is associated with increased residual risk that may benefit from aggressive or novel therapies.

contacting health.resdata@gov.ab.ca for researchers who meet the criteria for access to confidential information.

**Funding:** The authors received no specific funding for this work.

**Competing interests:** No authors have competing interests.

## Introduction

Advances in the treatment of acute coronary syndromes (ACS) over the past several decades have resulted in significant improvements in clinical outcomes [1, 2]. Despite this, a substantial proportion of individuals will experience a subsequent cardiovascular (CV) event [3]. This residual risk is likely attributed to inflammatory, prothrombotic and metabolic pathways that are currently not effectively addressed by available therapies and is influenced by concomitant comorbidities [4, 5]. Estimates of residual risk of further cardiovascular events have traditionally been derived from large clinical trials, however these estimates are limited by the lack of generalizability to a 'real world' population and a relatively low number of clinical events. As a result, there has been significant interest in estimating residual cardiovascular risk in large population-based studies and determining the impact of commonly associated comorbidities, such as heart failure [6], hypertension [7], peripheral vascular disease [8], renal disease [6, 9], cerebrovascular disease [8], diabetes [6] and others [10]. Our study aimed to determine the impact of these comorbidities on residual CV risk in a large, inclusive population of patients following an ACS.

## Methods

### Study design and data sources

This was a retrospective, population level study of patients with an ACS hospitalisation between April 1st, 2010 and March 31st, 2016 in Alberta, Canada. ACS admissions were identified using international classification of disease codes (S1 Table) in a linked administrative database that has previously been described [11]. Briefly, a unique personal identification number was used to link the following: hospitalization records containing data on a main and 24 secondary diagnoses, procedures or interventions; physician claims to assess post-discharge follow-up care; emergency department and outpatient clinic records; pharmaceutical claims to assess prescribed medications, and the Alberta Health Care Insurance Plan (AHICP) which provides patient demographic data. These administrative health databases were linked to a cardiac catheterization (the Alberta Provincial Project for Outcome Assessment in Coronary Heart Disease (APPROACH) registry for information on coronary anatomy for patients who underwent coronary angiography.

Hospitalizations of the same patient occurring within 24 h were clustered into episodes of care. Non-residents of Alberta, patients aged 17 years or less at discharge, patients with intra-hospital transfer following an initial hospitalization for non-ACS diagnosis, patients with diagnosis of unspecified myocardial infarction, patients with ACS preceding within 90 days of the index event, and those with an angiography within 180 days of the index event were excluded from the study. Recent angiography and ACS patients were excluded to avoid patients presenting for staged procedures and periprocedural MIs. For each patient, only the earliest episode during the study period was included.

Ethical approval was received from the University of Alberta Health Research Ethics Board (Pro00062982). The ethics panel determined that the research is a retrospective database review for which subject consent for access to personally identifiable health information would not be reasonable, feasible, or practical.

### Definition of outcomes and covariates

The primary outcome of the study was the time to the earliest of major adverse cardiac events (MACE), that included death from any cause, hospitalization for stroke as any diagnosis following the index admission, or re-hospitalization following discharge for myocardial

infarction (MI) as any diagnosis. The secondary outcomes were the individual components of the composite endpoint: death from any cause, hospitalization for stroke as any diagnosis following the index hospital admission for ACS or re-hospitalization following discharge for MI as any diagnosis. The date of MACE outcome was taken as the earliest of the death date, admission date of a hospitalization for stroke or MI. For each outcome the time from index ACS admission to outcome, death, first out-of-province migration date as shown by AHICP registry or lost to follow up on March 31st, 2016 was calculated. For the outcomes of stroke and recurrent MI, patients without events were censored at the date of death, first out-of-province migration date or March 31st, 2016, whichever came first.

Prior myocardial infarction was assessed from all diagnosis fields of all hospitalizations discharged within one year prior to index ACS episode admission. Cerebrovascular disease was assessed from all diagnosis fields of all hospitalizations discharged within one year prior to or within the index ACS episode, excluding instances where I63, ICD-10 code for stroke, was identified within the index episode. Stroke events identified at index were considered outcomes rather than comorbidities. Remaining comorbidities were assessed from all secondary diagnosis fields of all hospitalizations during the index ACS episode using ICD codes (S2 Table). A comorbidity count variable (none, one, two, three or more) was constructed based on the presence or absence of heart failure, diabetes, renal disease, peripheral vascular disease, cerebrovascular disease and hypertension. Coronary angiography at index, percutaneous coronary intervention (PCI) and coronary artery bypass grafting (CABG) within 30 days after index admission were extracted from APPROACH data. Use of thrombolysis at index was extracted from all procedure codes of all index hospitalizations for ST elevation myocardial infarction patients, only. Comorbidities, procedures, and medications that were not identified in our data were assume as not being present.

## Statistical analysis

Patients were stratified according to the type of ACS: unstable angina (UA), non-ST elevation myocardial infarction (NSTEMI), and ST elevation myocardial infarction (STEMI). Characteristics of patients (age, sex, urban/rural residence, comorbidities, Charlson comorbidity score), of hospitals (tertiary/non-tertiary type, length of stay), ambulance arrival, and in-hospital mortality were compared across the three groups. In addition, patient management at index (coronary angiography, PCI, CABG) and medications within three months after discharge were compared across the three groups. Medications were assessed within the cohort of patients that were discharged alive and survived at least 3 months after index discharge. Categorical variables were presented using counts and percentages and compared using the $\chi^2$ test. Continuous variables were presented using mean and standard deviation and compared using one-way ANOVA, and using median and interquartile range and compared using Kruskal-Wallis test. Using Kaplan-Meier estimator, we constructed cumulative incidence (CI) curves separately for the MACE endpoint and death. These were stratified by type of ACS at index, and presence or absence of the following comorbidities at index ACS hospitalization: diabetes, renal disease, heart failure, peripheral vascular disease, cerebrovascular disease, and hypertension. In addition, CI curves for patients with one, two, or three or more comorbidities were constructed. CI curves were compared using log-rank test for time to event. Rates of MACE and secondary outcomes were calculated as the ratio of number experiencing the event over the sum of the time to MACE endpoint, and reported per 10,000 person-years.

Cox proportional hazard models for MACE and all-cause mortality, and Fine-Gray models for recurrent myocardial infraction and stroke with death as competing event, were used to examine the independent association between comorbidities of interest and outcomes. These

multivariable models included the following: type of ACS, sex, age (as a continuous variable), urban/rural residence, hypertension, heart failure, diabetes, peripheral vascular disease, cerebrovascular disease, renal disease, prior myocardial infarction, peptic ulcer disease, dementia, COPD, liver disease, cancer, connective tissue disease and hemiplegia. To estimate the effects of comorbidity count we included all previously listed covariates, replacing the six individual comorbidities by their count.

All statistical analyses were performed using SAS version 9.4.

## Results

### Patient population

A total of 43,353 hospitalizations for ACS were identified between April 2010 and March 2016 in the province of Alberta as part of 38,294 episodes of care (S1 Fig). Following exclusion of episodes of patients residing out of province (n = 265), aged 17 years or less at discharge (n = 3), episodes with intra-hospital transfers following an initial admission for non-ACS diagnosis (n = 1042), episodes for unspecified MI diagnosis (n = 1805), episodes with preceding ACS events within 90 days (n = 1090), with angiography within 180 days of the index event (n = 956), and non-index episodes (n = 2080) a total of 31,056 ACS hospitalization episodes of 31,056 patients remained and were included in the analysis.

The mean age was 66.2 years +/- 13.8 and 30.7% (n = 9529) were female (Table 1). Of these, 15,358 (49.5%) presented with a non-ST segment elevation myocardial infarction (NSTEMI), 10,563 (34.0%) with an ST segment elevation myocardial infarction and 5,135 (16.5%) with unstable angina (UA). Of the entire cohort, 11.1% (n = 3,451) had a previous history of heart failure, 63.0% (n = 19,566) hypertension, 28.3% (n = 8,774) diabetes, 2.4% (n = 751) cerebrovascular disease, 2.9% (n = 908) peripheral vascular disease and 4.0% (n = 1,239) renal disease. Of the total cohort, 41.2% (n = 12,792) had one of these comorbidities, 24.7% (n = 7,672) had two and 6.6% (n = 2,041) had three or more. The NSTEMI cohort was older (68.4 years +/- 14.1) compared to the STEMI (62.8 years +/- 13.4) and UA (66.6 years +/- 11.9) cohort.

### In-hospital management and outcomes

The mean hospital length of stay was 7.5 +/- 13.3 days, with the shortest length of stay seen in UA (5.8+/- 8.0 days) and the longest length of stay seen in NSTEMI (8.7 +/- 15.1 days). Overall, 81.6% underwent coronary angiography during the index hospitalisation, 57.7% underwent percutaneous coronary intervention (PCI) within 30 days of admission and 6.1% underwent coronary artery bypass grafting surgery (CABG) within 30 days of admission (S2 Table). Of patients presenting with a STEMI, 25.8% received fibrinolysis. Rates of angiography and PCI were higher in the STEMI population (92.0% and 80.6%) compared to patients presenting with an NSTEMI (76.5% and 47.7%) and UA (75.5% and 40.8%). However, patients presenting with an NSTEMI and UA had the highest rates of CABG (8.0% and 7.8%) compared to patients with a STEMI (2.5%). Patients presenting with a STEMI were more frequently discharged on evidenced based therapies including P2Y12 inhibitor, beta-blocker, angiotensin converting enzyme inhibitors and statins (S2 Table).

### Long-term outcomes

Over an average of 3.0 +/- 3.7 years of follow-up, the composite of death, recurrent myocardial infarction and stroke (Fig 1) was highest in the NSTEMI cohort (3.58 per 10,000 person-years) followed by the STEMI cohort (2.41 per 10,000 person-years) and UA cohort (1.68 per 10,000 person-years; Table 2). The presence of any of the comorbidities of interest was associated

**Table 1. Baseline demographics stratified by ACS type.**

| | UA | NSTEMI | STEMI | Total |
|---|---|---|---|---|
| Total N | 5135 | 15358 | 10563 | 31056 |
| Age, years mean (SD) | 66.6 (11.9) | 68.4 (14.1) | 62.8 (13.4) | 66.2 (13.8) |
| median (IQR) | 66.0 (58.0, 75.0) | 68.0 (58.0, 80.0) | 61.0 (53.0, 72.0) | 65.0 (56.0, 77.0) |
| Female sex | 1613 (31.4) | 5224 (34.0) | 2692 (25.5) | 9529 (30.7) |
| Male sex | 3522 (68.6) | 10134 (66.0) | 7871 (74.5) | 21527 (69.3) |
| Urban residence | 4138 (80.6) | 12167 (79.2) | 8529 (80.7) | 24834 (80.0) |
| Arrival at tertiary hospital | 1584 (30.8) | 4579 (29.8) | 8067 (76.4) | 14230 (45.8) |
| Arrival by ambulance | 1620 (31.5) | 6880 (44.8) | 6341 (60.0) | 14841 (47.8) |
| Heart failure | 275 (5.4) | 2133 (13.9) | 1043 (9.9) | 3451 (11.1) |
| Hypertension | 3659 (71.3) | 10240 (66.7) | 5667 (53.6) | 19566 (63.0) |
| Diabetes | 1629 (31.7) | 4785 (31.2) | 2360 (22.3) | 8774 (28.3) |
| Atrial fibrillation/flutter | 481 (9.4) | 1929 (12.6) | 912 (8.6) | 3322 (10.7) |
| Peripheral vascular disease | 173 (3.4) | 522 (3.4) | 213 (2.0) | 908 (2.9) |
| Cerebrovascular disease | 101 (2.0) | 483 (3.1) | 167 (1.6) | 751 (2.4) |
| Dementia | 40 (0.8) | 560 (3.6) | 160 (1.5) | 760 (2.4) |
| COPD | 429 (8.4) | 1777 (11.6) | 690 (6.5) | 2896 (9.3) |
| Renal disease | 169 (3.3) | 813 (5.3) | 257 (2.4) | 1239 (4.0) |
| Cancer | 82 (1.6) | 444 (2.9) | 210 (2.0) | 736 (2.4) |
| Charlson score, mean (SD) | 1.1 (1.4) | 1.4 (1.6) | 0.9 (1.4) | 1.2 (1.5) |
| median (IQR) | 1.0 (0.0, 2.0) | 1.0 (0.0, 2.0) | 0.0 (0.0, 2.0) | 0.0 (0.0, 2.0) |
| Death in hospital | 21 (0.4) | 613 (4.0) | 598 (5.7) | 1232 (4.0) |
| Episode length of stay, day, mean (SD) | 5.8 (8.0) | 8.7 (15.1) | 6.7 (12.4) | 7.5 (13.3) |
| median (IQR) | 4.0 (2.0, 6.0) | 5.0 (3.0, 9.0) | 4.0 (3.0, 6.0) | 4.0 (3.0, 7.0) |

All comparisons were significant at p<0.01.

Abbreviations: ACS—acute coronary syndromes; UA—unstable angina; NSTEMI—non-ST segment elevation myocardial infarction; STEMI—ST segment elevation myocardial infarction; COPD—chronic obstructive pulmonary disease; SD—standard deviation; IQR—interquartile range

with a higher rate of the primary composite outcome (Fig 2) and mortality (S1 Fig). Heart failure was associated with the highest risk for the composite outcome [adjusted hazard ration (aHR) 1.8; 95% CI 1.7–1.9], followed by renal disease (aHR 1.5; 95% CI 1.4–1.7), diabetes (aHR 1.5; 95% CI 1.4–1.6), peripheral vascular disease (aHR 1.3; 95% CI 1.2–1.4), and

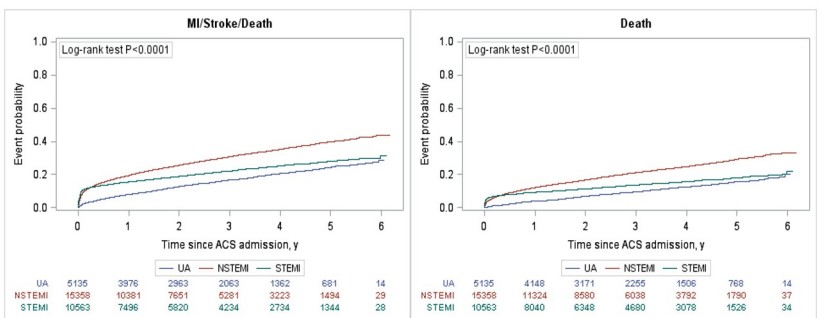

**Fig 1. Kaplan Meyer curves of MACE and death stratified by ACS type.** ACS—acute coronary syndromes; UA—unstable angina; NSTEMI—non-ST segment elevation myocardial infarction; STEMI—ST segment elevation myocardial infarction; MI—myocardial infarction.

**Table 2. Long-term outcomes following discharge stratified by ACS type and presence of comorbidities.**

| | Composite | | | Death | | | Stroke | | | MI | | |
|---|---|---|---|---|---|---|---|---|---|---|---|---|
| | Rate* | aHR (95%CI)** | P | Rate | aHR (95%CI) | P | Rate | aHR (95%CI) | P | Rate | aHR (95%CI) | P |
| UA (n = 5135) | 1.68 | 0.57 (0.53, 0.62) | <.0001 | 0.95 | 0.57 (0.51, 0.63) | <.0001 | 0.17 | 0.72 (0.56, 0.93) | 0.011 | 0.79 | 0.72 (0.64, 0.81) | <.0001 |
| NSTEMI (n = 15358) | 3.58 | 0.95 (0.90, 1.00) | 0.059 | 2.23 | 0.93 (0.87, 0.99) | 0.0273 | 0.27 | 0.91 (0.76, 1.09) | 0.3015 | 1.55 | 1.20 (1.11, 1.30) | <.0001 |
| STEMI (n = 10563) | 2.41 | 1 | | 1.38 | 1 | | 0.2 | 1 | | 0.98 | 1 | |
| No Heart failure (n = 27605) | 2.31 | 1 | | 1.28 | 1 | | 0.19 | 1 | | 1.08 | 1 | |
| Heart failure (n = 3451) | 8.91 | 1.83 (1.73, 1.93) | <.0001 | 6.38 | 2.13 (1.99, 2.26) | <.0001 | 0.6 | 1.37 (1.11, 1.69) | 0.0028 | 2.82 | 1.26 (1.15, 1.39) | <.0001 |
| No Hypertension (n = 11490) | 2.48 | 1 | | 1.58 | 1 | | 0.14 | 1 | | 0.99 | 1 | |
| Hypertension (n = 19566) | 3.02 | 0.88 (0.84, 0.93) | <.0001 | 1.78 | 0.78 (0.73, 0.82) | <.0001 | 0.28 | 1.53 (1.27, 1.83) | <.0001 | 1.35 | 1.08 (1.00, 1.17) | 0.0398 |
| No Diabetes (n = 22282) | 2.39 | | | 1.45 | | | 0.19 | 1 | | 1 | | |
| Diabetes (n = 8774) | 4.08 | 1.51 (1.44, 1.59) | <.0001 | 2.41 | 1.56 (1.47, 1.66) | <.0001 | 0.32 | 1.30 (1.10, 1.53) | 0.0017 | 1.84 | 1.56 (1.46, 1.68) | <.0001 |
| No CVD (n = 30305) | 2.73 | 1 | | 1.64 | 1 | | 0.21 | 1 | | 1.2 | 1 | |
| CVD (n = 751) | 7.54 | 1.27 (1.14, 1.41) | <.0001 | 4.93 | 1.30 (1.15, 1.47) | <.0001 | 1.12 | 1.94 (1.34, 2.82) | 0.0005 | 2.04 | 0.97 (0.80, 1.17) | 0.7316 |
| No PVD(n = 30148) | 2.74 | 1 | | 1.65 | 1 | | 0.21 | 1 | | 1.19 | 1 | |
| PVD (n = 908) | 5.83 | 1.28 (1.16, 1.42) | <.0001 | 3.56 | 1.26 (1.11, 1.42) | 0.0002 | 0.69 | 1.98 (1.48, 2.65) | <.0001 | 2.24 | 1.24 (1.05, 1.46) | 0.0094 |
| No Renal disease (n = 29817) | 2.64 | 1 | | 1.56 | 1 | | 0.22 | 1 | | 1.15 | 1 | |
| Renal disease (n = 1239) | 8.86 | 1.52 (1.40, 1.65) | <.0001 | 6.15 | 1.60 (1.46, 1.75) | <.0001 | 0.45 | 0.84 (0.60, 1.18) | 0.3242 | 3.16 | 1.43 (1.24, 1.63) | <.0001 |
| Comorbidities count, 0 (n = 8551) | 1.7 | 1 | | 0.99 | 1 | | 0.08 | 1 | | 0.78 | 1 | |
| 1 (n = 12792) | 2.24 | 1.05 (0.99, 1.13) | 0.1144 | 1.29 | 0.97 (0.89, 1.05) | 0.4469 | 0.17 | 1.93 (1.45, 2.55) | <.0001 | 1 | 1.14 (1.03, 1.26) | 0.0091 |
| 2 (n = 7672) | 3.98 | 1.57 (1.47, 1.68) | <.0001 | 2.37 | 1.48 (1.36, 1.61) | <.0001 | 0.32 | 2.95 (2.21, 3.95) | <.0001 | 1.72 | 1.65 (1.49, 1.83) | <.0001 |
| >= 3 (n = 2041) | 9.98 | 2.61 (2.40, 2.85) | <.0001 | 6.21 | 2.55 (2.31, 2.82) | <.0001 | 1.2 | 6.66 (4.78, 9.29) | <.0001 | 3.43 | 2.26 (1.97, 2.59) | <.0001 |

*Rate = per 10000 person years

** Models included the following covariates: type of ACS, sex, continuous age, urban/rural residence, hypertension, prior myocardial infarction, heart failure, diabetes, peptic ulcer disease, peripheral vascular disease, cerebrovascular disease, dementia, COPD, renal disease, liver disease, cancer, connective tissue disease, hemiplegia. Models used to estimate the effect of comorbidity count included all previously listed covariates, replacing the six individual comorbidities by their count.

Abbreviations: HR—hazard ratio; CI—confidence interval; UA—unstable angina; NSTEMI—non-ST segment elevation myocardial infarction; STEMI—ST segment elevation myocardial infarction; CVD—cerebrovascular disease; PVD—peripheral vascular disease

cerebrovascular disease (aHR 1.3; 95% CI 1.1–1.4). Hypertension was not found to be associated with an increase in MACE after adjustment (aHR 0.88; 95% CI 0.8–0.9). There was a clear comorbidity-response effect: compared to ACS patients with none of the comorbidities of interest, having one, two, or three+ comorbidities were associated with a 10% (aHR of 1.1; 95% CI 0.99–1.1), 60% (aHR 1.57; 95% CI 1.5–1.7) and almost three times (aHR 2.6; 95% CI 2.4–2.9) increase in the risk of the composite endpoint (Fig 3), respectively. Similar trends were observed for individual components of the composite endpoint (Table 2).

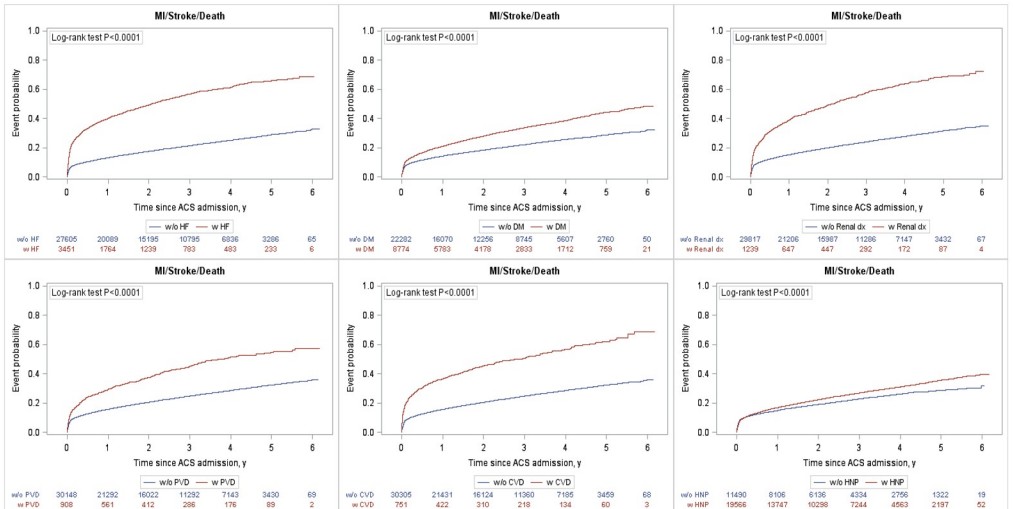

**Fig 2. Kaplan Meyer curves of MACE stratified by the presence or absence of heart failure, diabetes, renal disease, peripheral vascular disease, cerebrovascular disease and hypertension.** ACS—acute coronary syndromes; UA—unstable angina; NSTEMI—non-ST segment elevation myocardial infarction; STEMI—ST segment elevation myocardial infarction; MI—myocardial infarction; HF—heart failure; DM—diabetes; dx—disease; PVD—peripheral vascular disease; CVD—cerebrovascular disease; HNP—hypertension.

## Discussion

Despite advances in the management of ACS, there still continues to be significant risk of recurrent cardiovascular events following an index ACS presentation. The presence of common pre-existing comorbidities, such as heart failure, diabetes and peripheral vascular disease is associated with significantly worse long-term outcomes. Individuals with an NSTEMI demonstrated the highest risk, with ~40% of individuals experiencing a component of the endpoint by the 5-year mark. While this may appear unexpectedly high compared to clinical trial data, trials have been shown to demonstrate selection bias towards younger and healthier participants, leading to an underestimation of prognosis compared to inclusive, population level observational data [12]. Studies exploring long-term outcomes following ACS have been limited by small sample sizes, dated data or incomplete reporting of outcomes of interest [13]. Contemporary, generalizable estimates of residual cardiovascular risk have recently been reported from the Reduction of Atherothrombosis for Continued Health (REACH) international registry which included 16,770 patients with a prior history of myocardial infarction

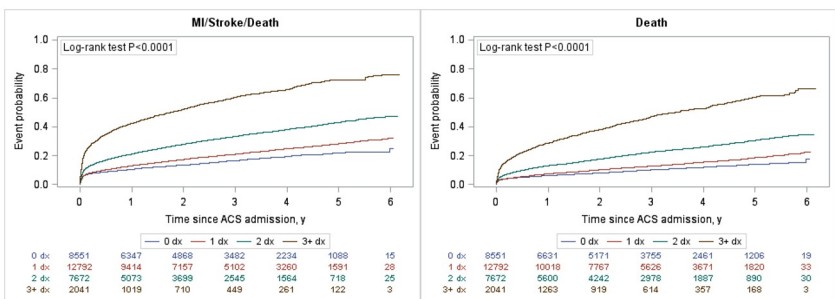

**Fig 3. Kaplan Meyer curves of MACE and death stratified by presence of 1, 2 or 3+ comorbidities examined.** ACS—acute coronary syndromes; MI—myocardial infarction; dx—disease.

[14]. Over a 4-year period, the incidence of any of cardiovascular death, recurrent myocardial infarction and stroke was 15%, considerably lower than our findings. Reasons for this many include that the REACH analysis is likely driven by the timing of recruitment. More than 75% of patients in the REACH registry were enrolled >1 year since their ACS presentation, likely leading to an underestimating of cardiovascular complications which is thought to be higher in the period immediately after an ACS.

As the acute management of ACS continues to improve, older and more comorbid individuals are surviving past their initial ACS presentation which was demonstrated in an administrative database of >6.5 million ACS presentations, where over a 11-year period comorbidity burden of patients with ACS increased significantly [10]. In their analysis, each incremental pre-existing comorbidity was associated with worse in-hospital outcomes. Additional studies have demonstrated that the adverse impact of pre-existing comorbidities is observed up to 1-year following their index event [15].

Our study has several limitations that should be considered, many of which are universal to studies using administrative data. These include the inability to adjudicate clinical events, outcomes and covariates and susceptibility to coding error. Identification of MI or stroke outcomes are specifically susceptible to either over or under representation depending on coding definitions. By limiting the definition to the primary diagnosis field there runs the risk of underestimation of outcomes and inversely the use of all diagnosis fields runs the risk of overestimation. In our analysis, 60.5% of the primary outcome were identified from the primary diagnosis field. Additionally, administrative data is limited by the inability to ascertain adherence to medical therapies and inability to adjudicate severity of comorbidities and clinical endpoints. For example, aspirin use is inaccurately captured in our database as over-the-counter medications are not documented in the pharmaceutical information network linked to our database.

In conclusion, in a large contemporary and generalizable population we have demonstrated that there is a significant risk of recurrent cardiovascular events following an index ACS presentation. The incremental presence of commonly observed comorbidities is associated with worse long-term outcomes. These individuals have the highest residual risk for a subsequent event and represent the greatest opportunity for novel interventions to demonstrate a meaningful clinical benefit and cost-effectiveness.

## Supporting information

**S1 Table. Variable definitions.** MI—myocardial infarction, NSTEMI—non-ST segment elevation myocardial infarction; STEMI—ST segment elevation myocardial infarction; MI—myocardial infarction; HF—heart failure; PVD- peripheral vascular disease; CVD—cerebrovascular disease; DAD—discharge abstract database; dx—diagnosis.
(DOCX)

**S2 Table. In-hospital management of ACS.** ACS—acute coronary syndromes; UA—unstable angina; NSTEMI—non-ST segment elevation myocardial infarction; STEMI—ST segment elevation myocardial infarction; CABG—coronary artery bypass grafting; PCI—percutaneous coronary intervention; ACEi—angiotensin converting enzyme inhibitor; ARB: angiotensin receptor blocker; MRA—mineralocorticoid receptor antagonist/aldosterone receptor antagonist.
(DOCX)

**S1 Fig. Inclusion criteria.**
(TIF)

**S2 Fig. Kaplan Meyer curves of death stratified by the presence or absence of heart failure, diabetes, renal disease, peripheral vascular disease, cerebrovascular disease and hypertension.**
(TIF)

## Author Contributions

**Conceptualization:** Pishoy Gouda, Kevin R. Bainey, Padma Kaul, Robert C. Welsh.

**Data curation:** Padma Kaul, Robert C. Welsh.

**Formal analysis:** Pishoy Gouda, Anamaria Savu, Padma Kaul.

**Methodology:** Pishoy Gouda, Anamaria Savu, Kevin R. Bainey, Padma Kaul, Robert C. Welsh.

**Project administration:** Robert C. Welsh.

**Resources:** Robert C. Welsh.

**Supervision:** Robert C. Welsh.

**Writing – original draft:** Pishoy Gouda, Robert C. Welsh.

**Writing – review & editing:** Pishoy Gouda, Anamaria Savu, Kevin R. Bainey, Padma Kaul, Robert C. Welsh.

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
