## [Decision Letter · Decision Letter 0]

20 Apr 2021

PONE-D-21-04254

Long-term Risk of Death and Recurrent Cardiovascular Events Following Acute Coronary Syndromes

PLOS ONE

Dear Dr. Welsh,

Thank you for submitting your manuscript to PLOS ONE. After careful consideration, we feel that it has merit but does not fully meet PLOS ONE’s publication criteria as it currently stands. Therefore, we invite you to submit a revised version of the manuscript that addresses the points raised during the review process.

We look forward to receiving your revised manuscript.

Kind regards,

Yoshihiro Fukumoto

Academic Editor

PLOS ONE

Journal Requirements:

Please provide additional details regarding participant consent. In the ethics statement in the Methods and online submission information, please ensure that you have specified (1) whether consent was informed and (2) what type you obtained (for instance, written or verbal, and if verbal, how it was documented and witnessed). If your study included minors, state whether you obtained consent from parents or guardians. If the need for consent was waived by the ethics committee, please include this information.

We note that you have indicated that data from this study are available upon request. PLOS only allows data to be available upon request if there are legal or ethical restrictions on sharing data publicly. For information on unacceptable data access restrictions, please see http://journals.plos.org/plosone/s/data-availability#loc-unacceptable-data-access-restrictions.

3a) If there are ethical or legal restrictions on sharing a de-identified data set, please explain them in detail (e.g., data contain potentially identifying or sensitive patient information) and who has imposed them (e.g., an ethics committee). Please also provide contact information for a data access committee, ethics committee, or other institutional body to which data requests may be sent.

3b) If there are no restrictions, please upload the minimal anonymized data set necessary to replicate your study findings as either Supporting Information files or to a stable, public repository and provide us with the relevant URLs, DOIs, or accession numbers. Please see http://www.bmj.com/content/340/bmj.c181.long for guidelines on how to de-identify and prepare clinical data for publication. For a list of acceptable repositories, please see http://journals.plos.org/plosone/s/data-availability#loc-recommended-repositories.

Reviewers' comments:

Reviewer's Responses to Questions

**Comments to the Author**

1. Is the manuscript technically sound, and do the data support the conclusions?

Reviewer #1: Yes

Reviewer #2: Yes

2. Has the statistical analysis been performed appropriately and rigorously? 

Reviewer #1: I Don't Know

Reviewer #2: Yes

3. Have the authors made all data underlying the findings in their manuscript fully available?

Reviewer #1: Yes

Reviewer #2: Yes

4. Is the manuscript presented in an intelligible fashion and written in standard English?

Reviewer #1: Yes

Reviewer #2: Yes

5. Review Comments to the Author

Reviewer #1: The authors tried to show the long-term cardiovascular (CV) event risk in patients with acute coronary syndrome (ACS) and the factors that were associated with the risk. The results indicated that patients with NSTEMI, rather than those with STEMI or unstable angina, were at a higher risk for major adverse cardiac events (MACE) defined by a composite of all-cause death, hospitalization for stroke, and re-hospitalization for MI. The incremental presence of comorbidities was associated with worse long-term outcomes in the study patients with ACS. Thus, the authors’ observations agreed with previous reports and added valuable data.

The strength of the current study is the large number of study patients and the data of mortality might be accurate, while the weaknesses of the study include the lack of information about the status of risk management, adherence to the medical therapies and detailed information (such as severity) of other endpoints and comorbidities.

Because the study patients were those with ACS presentation and diabetes and hypertension were included in the comorbidity count variables, another common risk factor for coronary artery disease, dyslipidemia or hyper-LDL cholesterolemia should be included in the variables as well. Why the authors excluded this factor from the comorbidity count variables? If the reason was that hyper-LDL cholesterolemia was not a significant risk factor for future CV events in this study, the use of statins in patients with hyper-LDL cholesterolemia and non-use in those without hyper-LDL cholesterolemia might cancel the significance of this risk factor. Thus, the lack of information of control status of each risk factor could affect the results. Please consider adding hyper-LDL cholesterolemia to the comorbidity count variables, if the data are available.

Use of the term “residual risk” in the current study seems somewhat strange to me. Residual risks may be defined as risks remained after intensive or at least standard risk control was achieved, but the risk management status was uncertain.

Reviewer #2: This is an interesting study reporting the impact of the comorbidities on residual CV risk in a large, inclusive population of patients following an ACS. The sample size is large. The paper is well-written. I have only one comment to the authors.

What is the novelty of this study? I think several reports have already shown clinical significance of the comorbidities in ACS patients. Could the authors highlight novelty of this study in discussion?

6. PLOS authors have the option to publish the peer review history of their article (what does this mean?). If published, this will include your full peer review and any attached files.

Reviewer #1: No

Reviewer #2: No

---

## [Author Response · Author response to Decision Letter 0]

12 Jun 2021

Response to Reviewers

Reviewer #1

Comment # 1

The authors tried to show the long-term cardiovascular (CV) event risk in patients with acute coronary syndrome (ACS) and the factors that were associated with the risk. The results indicated that patients with NSTEMI, rather than those with STEMI or unstable angina, were at a higher risk for major adverse cardiac events (MACE) defined by a composite of all-cause death, hospitalization for stroke, and re-hospitalization for MI. The incremental presence of comorbidities was associated with worse long-term outcomes in the study patients with ACS. Thus, the authors’ observations agreed with previous reports and added valuable data.

Response to comment #1 - Thank you

Comment #2

The strength of the current study is the large number of study patients and the data of mortality might be accurate, while the weaknesses of the study include the lack of information about the status of risk management, adherence to the medical therapies and detailed information (such as severity) of other endpoints and comorbidities.

Response to comment #2 – These limitations of administrative data have been added to the limitations section of the discussion.

“Additionally, administrative data is limited by the inability to ascertain adherence to medical therapies and inability to adjudicate severity of comorbidities and clinical endpoints.”

Comment #3

Because the study patients were those with ACS presentation and diabetes and hypertension were included in the comorbidity count variables, another common risk factor for coronary artery disease, dyslipidemia or hyper-LDL cholesterolemia should be included in the variables as well. Why the authors excluded this factor from the comorbidity count variables? If the reason was that hyper-LDL cholesterolemia was not a significant risk factor for future CV events in this study, the use of statins in patients with hyper-LDL cholesterolemia and non-use in those without hyper-LDL cholesterolemia might cancel the significance of this risk factor. Thus, the lack of information of control status of each risk factor could affect the results. Please consider adding hyper-LDL cholesterolemia to the comorbidity count variables, if the data are available.

Response to comment #3

Thank you for your comments. While we agree that dyslipidemia is a common risk factor for coronary artery disease, its use in administrative databases is challenging. As per your suggestion, we explored ICD-10 codes for dyslipidemia (E780-785). As you can see below, dyslipidemia was actually protective for our primary outcomes. We believe that is explained by the fact that patients that are diagnosed with dyslipidemia are also treated for it, and as a result have lower residual risk. As such, we have not included dyslipidemia in our comorbidity counts. 

Comment #4

Use of the term “residual risk” in the current study seems somewhat strange to me. Residual risks may be defined as risks remained after intensive or at least standard risk control was achieved, but the risk management status was uncertain.

Response to comment #4 - The term residual cardiovascular risk has been introduced recently in the acute coronary syndrome literature to describe the phenomenon where after an acute coronary syndrome has been treated, despite standard treatment an increased risk of secondary cardiovascular events remains. In our population, all were admitted to hospital, >80% underwent coronary angiography, ~64% underwent revascularization, which we describe as standard risk control in a generalized population. As such, we believe that our study findings should be labelled as residual risk as it pertains to a generalizable population.

Reviewer #2

Comment #5 This is an interesting study reporting the impact of the comorbidities on residual CV risk in a large, inclusive population of patients following an ACS. The sample size is large. The paper is well-written. I have only one comment to the authors. What is the novelty of this study? I think several reports have already shown clinical significance of the comorbidities in ACS patients. Could the authors highlight novelty of this study in discussion?

Response to comment #5 - 

Thank you for your review and insight. We believe that the greatest contribution of our manuscript is that it provides point estimates of the risk for secondary cardiovascular events following an ACS in a truly generalizable population, using recent data. Previous studies that describe this are frequently derived from clinical trial data, which has been observed to not fully represent the general ACS population that are seen in clinical practice. As observed in our data, our primary outcomes of myocardial infarction, stroke or death occurred in 40% of participants by the 5-year mark, highlighting the stark differences in comparing estimates from clinical trial participants and a more generalizable population. Secondly, observational data regarding this is either dated or includes smaller sample sizes. We believe that our data provides a benchmark for what the current residual risk is during our time period to provide a benchmark for future studies which aim to reduce residual risk. This is now increasingly highlighted in our discussion.

“While this may appear unexpectedly high compared to clinical trial data, trials have been shown to demonstrate selection bias towards younger and healthier participants, leading to an underestimation of prognosis compared to inclusive, population level observational data (12). Studies exploring long-term outcomes following ACS have been limited by small sample sizes, dated data or incomplete reporting of outcomes of interest (13).”

Journal Requirement Comments

Comment # 6 Please ensure that your manuscript meets PLOS ONE's style requirements, including those for file naming. The PLOS ONE style templates can be found at

Response to comment # 6 – We have reviewed the manuscript formatting guidelines and have made the required changes.

Comment # 7 Please provide additional details regarding participant consent. In the ethics statement in the Methods and online submission information, please ensure that you have specified (1) whether consent was informed and (2) what type you obtained (for instance, written or verbal, and if verbal, how it was documented and witnessed). If your study included minors, state whether you obtained consent from parents or guardians. If the need for consent was waived by the ethics committee, please include this information. If you are reporting a retrospective study of medical records or archived samples, please ensure that you have discussed whether all data were fully anonymized before you accessed them and/or whether the IRB or ethics committee waived the requirement for informed consent. If patients provided informed written consent to have data from their medical records used in research, please include this information.

Response to comment # 7 - The following statement has been added to the methods section of our manuscript.

“The ethics panel determined that the research is a retrospective database review for which subject consent for access to personally identifiable health information would not be reasonable, feasible, or practical.”  

Comment # 8 We note that you have indicated that data from this study are available upon request. PLOS only allows data to be available upon request if there are legal or ethical restrictions on sharing data publicly. For information on unacceptable data access restrictions, please see http://journals.plos.org/plosone/s/data-availability#loc-unacceptable-data-access-restrictions.

Response to comment # 8 - The data underlying this article was provided by the Government of Alberta under the terms of a research agreement. Inquiries respecting access to the data can be made to health.resdata@gov.ab.ca.

---

## [Decision Letter · Decision Letter 1]

18 Jun 2021

Long-term Risk of Death and Recurrent Cardiovascular Events Following Acute Coronary Syndromes

PONE-D-21-04254R1

Dear Dr. Welsh,

We’re pleased to inform you that your manuscript has been judged scientifically suitable for publication and will be formally accepted for publication once it meets all outstanding technical requirements.

Kind regards,

Yoshihiro Fukumoto

Academic Editor

PLOS ONE

Additional Editor Comments (optional):

Reviewers' comments:

Reviewer's Responses to Questions

**Comments to the Author**

1. If the authors have adequately addressed your comments raised in a previous round of review and you feel that this manuscript is now acceptable for publication, you may indicate that here to bypass the “Comments to the Author” section, enter your conflict of interest statement in the “Confidential to Editor” section, and submit your "Accept" recommendation.

Reviewer #1: All comments have been addressed

Reviewer #2: All comments have been addressed

2. Is the manuscript technically sound, and do the data support the conclusions?

Reviewer #1: Yes

Reviewer #2: Yes

3. Has the statistical analysis been performed appropriately and rigorously? 

Reviewer #1: I Don't Know

Reviewer #2: Yes

4. Have the authors made all data underlying the findings in their manuscript fully available?

Reviewer #1: Yes

Reviewer #2: Yes

5. Is the manuscript presented in an intelligible fashion and written in standard English?

Reviewer #1: Yes

Reviewer #2: Yes

6. Review Comments to the Author

Reviewer #1: The authors adequately addressed my comments and described difficult issues to be revised as study limitations. I have no further comments.

Reviewer #2: (No Response)

7. PLOS authors have the option to publish the peer review history of their article (what does this mean?). If published, this will include your full peer review and any attached files.

Reviewer #1: No

Reviewer #2: No

---

## [Editor Report · Acceptance letter]

24 Jun 2021

PONE-D-21-04254R1 

Long-term risk of death and recurrent cardiovascular events following acute coronary syndromes 

Dear Dr. Welsh:

I'm pleased to inform you that your manuscript has been deemed suitable for publication in PLOS ONE. Congratulations! Your manuscript is now with our production department. 

Kind regards, 

on behalf of

Dr. Yoshihiro Fukumoto 

Academic Editor

PLOS ONE